# Clinical Significance of the Fetuin-A-to-Adiponectin Ratio in Obese Children and Adolescents with Diabetes Mellitus

**DOI:** 10.3390/children8121155

**Published:** 2021-12-08

**Authors:** Moon-Bae Ahn, Seul-Ki Kim, Shin-Hee Kim, Won-Kyoung Cho, Jin-Soon Suh, Kyoung-Soon Cho, Byung-Kyu Suh, Min-Ho Jung

**Affiliations:** Department of Pediatrics, College of Medicine, The Catholic University of Korea, Seoul 06591, Korea; mbahn@catholic.ac.kr (M.-B.A.); seulki12633@gmail.com (S.-K.K.); tigger1018@naver.com (S.-H.K.); wendy626@catholic.ac.kr (W.-K.C.); rebekahjs@hanmail.net (J.-S.S.); soon926@catholic.ac.kr (K.-S.C.); suhbk@catholic.ac.kr (B.-K.S.)

**Keywords:** fetuin-A, adiponectin, diabetes mellitus, pediatric obesity

## Abstract

Fetuin-A and adiponectin are inflammatory cytokines associated with obesity and insulin resistance. This study aimed to examine the fetuin-A-to-adiponectin ratio (FAR) in diabetic children and to determine the role of FAR. A total of 54 children and adolescents with diabetes mellitus (DM) and 44 controls aged 9–16 years were included in this study. Clinical characteristics, including plasma fetuin-A and adiponectin levels, were compared with respect to body mass index (BMI) and diabetes type. Of 98 children, 54.1% were obese, whereas 18.4% were obese and diabetic. FAR was higher in obese children with DM than in non-obese children and also in type 2 DM children than in type 1. FAR showed a stronger association with BMI than with fetuin-A and adiponectin individually, and its association was more prominent in diabetic children than in controls. BMI was a risk factor for increased FAR. Plasma fetuin-A was elevated in obese children, and its association with insulin resistance and β cell function seemed more prominent in diabetic children after adjustment for adiponectin. Thus, FAR could be a useful surrogate for the early detection of childhood metabolic complications in diabetic children, particularly those who are obese.

## 1. Introduction

Childhood diabetes is a global public health issue with an increasing incidence in most developed countries [1]. Type 1 diabetes mellitus (T1DM) accounts for the majority of childhood diabetes; nonetheless, the incidence of type 2 diabetes mellitus (T2DM) is growing. The prevalence of diabetes mellitus (DM) in children is continuing to increase and shifts toward younger age groups. Obese children are at a higher risk for future vascular complications [2]. Childhood obesity not only triggers insulin resistance (IR), which is a hallmark of T2DM, but also contributes to metabolic derangements in children with T1DM [3,4]. Although T1DM is a state of insulin deficiency, long-term administration of exogenous insulin can lead to IR, which, in turn, disrupts glycemic control and results in adverse health outcomes, such as cardiovascular disease, stroke, and depression [5]. β cell dysfunction worsens over time in children with T2DM and, together with weight gain, can accelerate early peripheral and hepatic IR [6]. As obesity in diabetic children can be harmful, the prevention and management of obesity are therefore critical right from the time of DM diagnosis.

Fetuin-A, also known as alpha-2-Heremans-Schmid glycoprotein, is a multifunctional hepatokine involved in various endocrine signaling pathways. It is mostly associated with IR formation, with its proinflammatory effect causing diverse metabolic alterations [7]. The insulin signal in the muscles and liver is interrupted by fetuin-A via the autophosphorylation of insulin receptors, leading to hyperinsulinism [8]. Previous studies have revealed that an elevation in fetuin-A levels is associated with an increased risk of developing numerous childhood metabolic consequences of IR, including T2DM, nonalcoholic fatty liver disease (NAFLD), polycystic ovary syndrome, and excessive fetal growth [7,8,9,10]. These findings indicate that fetuin-A may be involved in the pathogenesis of IR and obesity in children.

Adiponectin is a 30-kDa adipokine involved in a wide range of roles associated with vascular biology and metabolism [11]. Low-molecular-weight adiponectin is related to IR, whereas high-molecular-weight adiponectin is related to insulin sensitivity [10]. Hypoadiponectinemia is associated with childhood obesity; however, the causality is unclear [12]. Chronic inflammation and prolonged exposure to insulin have been recognized as potential triggers for decreasing adiponectin gene expression in adipocytes [10]. Hypoadiponectinemia has also been reported in children with T2DM; nevertheless, it seems more likely to be associated with the development of IR than to be directly involved in the pathogenesis of T2DM [13]. The release of fetuin-A has been reported to possibly decrease insulin sensitivity via the suppression of adiponectin production in adipose tissues [14].

The genes encoding human fetuin-A and adiponectin are located next to each other on chromosome 3q27, which was previously mapped as a T2DM and metabolic syndrome (MetS) susceptibility locus; however, they act oppositely [15]. The fetuin-A-to-adiponectin ratio (FAR) was first introduced by Ju et al. in 2017, and its association with MetS in adults was shown to be stronger than that of either of its components [16]. Although it has been hypothesized that the FAR could be a sensitive indicator of obese children and adolescents as in those who are diabetic, clinical evidence regarding the value of fetuin-A, adiponectin, and FAR in such a population is limited. Therefore, the present study aimed to investigate plasma fetuin-A and adiponectin levels in diabetic children and adolescents in relation to obesity and to determine the role of FAR as a future indicator of metabolic complications when obesity and diabetes coexist.

## 2. Materials and Methods

### 2.1. Study Subjects

This study was designed as a case-control study including 54 children and adolescents diagnosed with T1DM or T2DM and 44 controls aged 9–16 years who visited two separate institutions (Bucheon St. Mary’s Hospital, Buchoen, Korea and Seoul St. Mary’s Hospital, Seoul, Korea). Monogenic or glucocorticoid-induced DM were excluded. Furthermore, obese patients and those with diabetes-related comorbidities, including cardiovascular, gastroesophageal, pulmonary, and renal diseases, were excluded. The controls were those who visited the endocrinology clinic for the purpose of growth assessment; the recruitment criteria were based on the absence of infection, neoplastic, endocrine, renal, and psychiatric disorders, as well as any underlying infections. The Institutional Review Board of the Catholic University of Korea approved this study (XC21SIDI0080 25 June 2021) according to the principles embodied in the Declaration of Helsinki. Informed consent was obtained from all participants and their guardians.

### 2.2. Definition of Obesity, DM, and MetS

DM diagnosis was established when clinical manifestation of DM, such as polyuria, polydipsia, polyphagia, or weight loss, were present in combination with the following criteria: randomly checked plasma glucose above 200 mg/dL (criterion i), 8 h fasting plasma glucose level above 126 mg/dL (criterion ii), 2 h postprandial plasma glucose above 200 mg/dL (criterion iii), or glycosylated hemoglobin (HbA1c) level above 6.5% (criterion iv) [17]. T1DM was confirmed among patients who presented with at least one of the following anti-pancreatic autoantibodies: insulin autoantibodies, glutamic acid decarboxylase 65 autoantibodies, β cell-specific zinc transporter 8 autoantibodies, or tyrosine phosphatase-like insulinoma antigen 2. Patients who were not classified as T1DM were classified as T2DM.

Participants were divided according to the body mass index standard deviation score (BMI SDS) into obese, overweight, and normoweight subgroups. Obesity, overweight, normoweight, and underweight were defined as BMI SDS ≥ 95th percentile, BMI SDS ≥ 85th but <95th percentile, BMI SDS ≥ 5th but <85th percentile, and BMI SDS < 5th percentile, respectively, for children’s age and sex on a national growth chart [17,18]. MetS was determined by the presence of obesity, in addition to two of any of the following: fasting glucose level of ≥100 mg/dL, triglyceride (TG) level of ≥150 mg/dL, and high-density lipoprotein cholesterol (HDL-C) level of <40 mg/dL for males and females under 16 years and <50 mg/dL for females over 16 years [19].

### 2.3. Anthropometric and Laboratory Parameters

Height scale (Harpenden Stadiometer, Holtain^®^, Crymych, UK) and weight scale (Simple Weighing Scale, Cas^®^, Seoul, Korea) were measured simultaneously when samples were collected, and BMI was calculated (kg/m^2^). All height and weight measurements were converted to sex- and age-matched standard deviation scores (SDSs) by using the national growth chart [18].

Fasting plasma samples were collected when study participants visited the endocrinology clinic. Blood analysis included glucose, aspartate aminotransferase (AST), alanine aminotransferase (ALT), C-reactive protein (CRP), total cholesterol (TC), TG, HDL-C, low-density lipoprotein cholesterol (LDL-C), uric acid, HbA1c, C-peptide, and insulin testing. The homeostatic model assessment of β cell function (HOMA-β) and IR (HOMA-IR) was calculated as the following: HOMA-β = (360 × fasting insulin [mU/L])/(glucose [mg/dL]-63) and HOMA-IR = fasting insulin (mU/L) × glucose (mg/dL)/405 [20].

### 2.4. Fetuin-A and Adiponectin Measurements

Plasma samples were immediately centrifuged at 3000× *g* for 15 min at 4 °C (U-32012 Centrifuge, Boeco^®^, Hamburg, Germany) upon collection and stored at −80 °C. Plasma fetuin-A and adiponectin concentrations were determined by using enzyme-linked immunosorbent assays (Fetuin-A, Human Fetuin-A Immunoassay, R&D Systems^®^, Minneapolis, MN, USA; Adiponectin, Human Total Adiponectin/Acrp30 Immunoassay, R&D Systems^®^, Minneapolis, MN, USA) in accordance with the manufacturer’s instructions. The manufactural dilution factor was used while each sample ran in duplicate. The average coefficients of variation for plasma intra-assay (inter-assay) precision were 4.3% (8.0%) and 3.5% (6.5%) for fetuin-A and adiponectin, respectively. The FAR value was derived from fasting fetuin-A (ng/mL) divided by adiponectin (ng/mL).

### 2.5. Statistics

Descriptive variables were expressed as the median (interquartile range[IQR] of 25% and 75%) while normal distribution was determined by using the Shapiro–Wilk test. Comparisons between T1DM and T2DM were performed by using the Mann–Whitney U test, whereas comparisons between more than three subcategorized groups with respect to obesity and diabetes status were conducted using the Kruskal–Wallis test. Spearman’s rank correlation was used to examine the correlations between clinical data and fetuin-A, adiponectin, and FAR. Univariate and multivariate regression analyses were performed to estimate the associated factors for the increased FAR in diabetic children. Along with binomial logistic regression analysis, a receiver operating characteristic (ROC) curve and the area under the curve (AUC) were generated to determine the optimal value for FAR in diabetic children to predict MetS. Data analyses were carried out by SPSS version 24.0 (IBM Corp., Armonk, NY, USA).

## 3. Results

### 3.1. Characteristics of Study Participants

This case-control study included 54 children with DM and 44 control patients with a median age of 12.42 years (range: 9.56–16.17 years). Among those with DM, 31 children had T1DM (57.4%), whereas 23 children had T2DM (42.6%). The duration of known diabetes was 1.58 years (1.00, 4.13 years). There were 53 (54.1%) obese children and 18 (33.3%) diabetic children. The baseline clinical characteristics of the diabetic children and controls in relation to obesity are presented in Table 1.

The participants were divided into four subgroups according to the presence of obesity and DM, and laboratory measurements were compared (Table 2). Age, BMI SDS, CRP, glucose, ALT, uric acid, HbA1c, HOMA-IR, and FAR were significantly higher in obese diabetic children than in non-obese controls, whereas adiponectin levels were significantly lower (*p* < 0.05). Age, BMI SDS, CRP, glucose, AST, ALT, TG, uric acid, C-peptide, HOMA-IR, HOMA-β, and FAR were significantly higher in obese diabetic children than in non-obese diabetic children, whereas HDL-C and adiponectin levels were significantly lower (*p* < 0.05). Fetuin-A levels were significantly higher in obese controls than in non-obese controls and non-obese diabetic children (*p* < 0.05).

Adiponectin levels (*p* < 0.001) were significantly lower, whereas FAR (*p* < 0.001) was higher in children with T2DM than in both non-diabetic children and those with T1DM. However, fetuin-A levels (*p* = 0.124) did not differ among the three groups (Table 3).

### 3.2. Fetuin-A, Adiponectin, and FAR of Diabetic Children and Controls in Relation to BMI SDS

[Fig children-08-01155-g001] presents a comparison of fetuin-A, adiponectin, and FAR between controls and diabetic children subcategorized into three groups with respect to BMI SDS.

Fetuin-A levels were the highest in obese controls and lowest in normoweight controls (*p* = 0.025). The adiponectin level was the lowest in obese controls, which was lower than that in overweight controls (*p* = 0.011) and normal controls (*p* = 0.004). FAR was higher in obese controls than in overweight controls (*p* = 0.012) and normoweight controls (*p* < 0.001).

The adiponectin level was the lowest in obese children with DM and highest in normoweight children with DM. Contrarily, FAR was the highest in obese children with DM and lowest in normoweight children with DM. Both adiponectin (*p* < 0.001) and FAR (*p* < 0.001) were significantly different between obese and normoweight children with DM. Fetuin-A levels showed no significant differences among the three groups.

### 3.3. Association of Fetuin-A, Adiponectin, and FAR with Metabolic Parameters of Diabetic Children

Correlation analyses between fetuin-A, adiponectin, and FAR with metabolic parameters were performed in diabetic children and controls (Table 4).

Fetuin-A levels were positively correlated with BMI SDS, whereas adiponectin was negatively correlated with BMI SDS (*p* = 0.002 vs. *p* < 0.001), AST (*p* = 0.001 vs. *p* = 0.029), ALT (*p* = 0.001 vs. *p* < 0.001), and HOMA-IR (*p* = 0.003 vs. *p* < 0.001) in diabetic children. Additionally, adiponectin was negatively correlated with TG (*p* < 0.001), uric acid (*p* < 0.001), C-peptide (*p* < 0.001), and HOMA-β (*p* < 0.001). In diabetic children, FAR was correlated with the same parameters as in adiponectin but in the opposite directions, showing a positive correlation with BMI SDS (*p* < 0.001), AST (*p* = 0.004), ALT (*p* < 0.001), TG (*p* < 0.001), uric acid (*p* < 0.001), C-peptide (*p* < 0.001), HOMA-IR (*p* = 0.001), and HOMA-β (*p* = 0.019) but a negative correlation with HDL-C (*p* < 0.001).

Adiponectin was negatively correlated with BMI SDS, whereas FAR was positively correlated with BMI SDS (*p* = 0.002 vs. *p* < 0.001), ALT (*p* = 0.012 vs. *p* = 0.017), uric acid (*p* < 0.001 vs. *p* < 0.001), HbA1c (*p* = 0.002 vs. *p* = 0.002), C-peptide (*p* < 0.001 vs. *p* < 0.001), HOMA-IR (*p* = 0.002 vs. *p* = 0.002), and HOMA-β (*p* = 0.007 vs. *p* = 0.009) in controls. Fetuin-A levels in controls showed a positive correlation with BMI SDS only (*p* = 0.003).

FAR (ρ = 0.679) exhibited a stronger association with BMI SDS than with fetuin-A (ρ = 0.404) and adiponectin (ρ = −0.649) individually in either group of diabetic children or controls. In addition, the association of FAR with BMI SDS was stronger in diabetic children (ρ = 0.679) than in controls (ρ = 0.540).

Univariate analyses revealed that increased FAR was associated with BMI, AST, ALT, HDL-C, uric acid, and C-peptide in diabetic children and with age, BMI, AST, ALT, uric acid, and C-peptide in controls (Table 5). BMI remained a risk factor for increased FAR in the final multivariate analyses after adjustment for age and sex, and its regression coefficient (β) indicated a stronger association in diabetic children (β = 86.7, *p* < 0.001) than in controls (β = 49.6, *p* = 0.004).

### 3.4. The FAR Cut-off Predicting MetS among Diabetic Children

Using logistic regression, FAR was determined to be a significant risk factor for incident MetS among diabetic children after adjustment for sex and age (odds ratio = 1.003, 95% Confidence Interval (CI) = 1.001–1.005; *p* = 0.018). An ROC curve was generated to evaluate the diagnostic performance of FAR in predicting childhood MetS, and the AUC was calculated as 0.87 (*p =* 0.018). An FAR of 284.91 was identified as the best cut-off value for diagnosing childhood MetS in diabetic children, with a corresponding sensitivity and specificity of 0.25 and 0.94, respectively ([Fig children-08-01155-g002]).

## 4. Discussion

To the best of our knowledge, our study is the first to evaluate the association between fetuin-A and adiponectin in diabetic children and adolescents, particularly in those who are obese, and to determine the role of FAR in predicting metabolic complications. Although adequate pediatric reference intervals for fetuin-A and adiponectin still need to be established, several large case-cohort studies conducted on the adult population have shown their link to the development of obesity-related comorbidities, such as T2DM and cardiovascular disease [21,22,23]. Fetuin-A, adiponectin, and FAR were elevated in obese children, irrespective of the presence of diabetes, and were more elevated in obese controls than in normoweight controls. In addition, adiponectin and FAR were more elevated in obese diabetic children than in normoweight diabetic children as well as in children with T2DM than in those with T1DM. The correlation between FAR and BMI was stronger than the correlation of FAR and fetuin-A or adiponectin alone. BMI was a unique risk factor for increased FAR in both diabetic children and controls, whereas the relationship between FAR and BMI was more prominent in diabetic children than in controls. Furthermore, increased FAR was a potential risk factor for MetS in diabetic children. Our findings suggest that FAR could provide a more precise interpretation than each component and could be a more sensitive surrogate for future metabolic disorders in diabetic children, particularly those who are obese. The results of the present study are consistent with those of previous studies [16,24].

Obesity is a major eliciting factor of diabetes associated with IR; the release of adipose tissue-derived mediators, including proinflammatory cytokines, is triggered by obesity [25]. Fetuin-A plays a role in the pathogenesis of IR by inhibiting insulin receptor activity via proteolysis of the α-chain and constitutive activation of insulin receptor tyrosine kinase activity, leading to a breakdown in insulin cascade pathways; however, the specific mechanism remains unclear [26]. IR and obesity are inseparable factors in the development of T2DM, and several adult studies have shown that their concomitant presence leads to higher fetuin-A levels, which are more prominent in diabetic than in non-diabetic individuals [26]. On the other hand, our study showed that despite being under independent circumstances of IR, fetuin-A did not differ between groups of children with respect to either the presence or type of DM. Because the fetuin-A levels were higher in metabolically healthy obese children than in both non-obese controls and non-obese diabetic children, it is also speculated that childhood obesity, even in the absence of IR, may be associated with an elevation in fetuin-A levels. Nevertheless, the fetuin-A levels in healthy children did not gradually increase as BMI increased; however, there was a clear difference in fetuin-A levels between obese and normoweight children.

In contrast to fetuin-A, adiponectin, the major adipocytokine secreted by differentiated adipocytes, is characterized by its anti-inflammatory and anti-apoptotic effects, and acts as a homeostatic player to improve insulin sensitivity by regulating glucose and lipid metabolism [27]. As both fetuin-A and adiponectin are linked to the mechanism underlying obesity but function in an opposite direction, the fetuin-A values adjusted by adiponectin seem to exhibit a higher performance. Zhou et al. showed the causal relationship between fetuin-A and various metabolic parameters associated with MetS, which was not highlighted in our study population [24]. Nonetheless, the following findings were consistent with the results of previous studies [24]: FAR in diabetic children (i) was the highest in the entire population, (ii) had a stronger correlation with BMI, and (iii) acted as a MetS indicator compared to controls. While chronic inflammation-induced IR has long been shown to be responsible for triggering obesity, numerous obesity-triggering inflammatory biomarkers have been introduced, and an understanding of their importance has been highlighted. Although its routine clinical purpose remains to be elucidated until its specific role is addressed, FAR could have a prognostic value in childhood obesity linked to metabolic derangement.

NAFLD is also a common adverse consequence of childhood obesity, and the presence of hepatosteatosis in obese children is positively correlated with higher levels of circulating proinflammatory cytokines, including fetuin-A [28]. Moreover, fetal islets exhibit more glucose intolerance than adult islets when exposed to higher concentrations of fetuin-A, and pancreatic inflammation triggered by NAFLD-derived fetuin-A secretion may accelerate the progression of β cell failure in diabetic patients [29]. Pyziak-Skupien et al. reported the association of higher fetuin-A levels with clinical partial remission in children with newly diagnosed T1DM, and they suggested fetuin-A as a potential predictor of remission [30]. Although a liver ultrasound was not performed in the study participants, our results indicated higher AST and ALT levels in obese children than in non-obese children; furthermore, fetuin-A was positively correlated with liver enzymes, C-peptide, and HOMA-IR in diabetic children. On the other hand, FAR also showed a positive correlation with those parameters in all participants but exhibited a stronger correlation in diabetic children. Therefore, FAR may provide a better interpretation of metabolic dysfunction, particularly in diabetic children.

The present study has a few limitations that need to be addressed. First, this was an age-unmatched case-control study that was conducted on a small sample size within a cross-sectional design. Although a significant difference in FAR was observed based on BMI status, the number of children with MetS was not sufficient to accurately determine the association between MetS and FAR. Second, correlation and regression analyses were not performed according to the type of diabetes despite their different auxological and laboratory characteristics. Third, considering the validity of FAR susceptibility to MetS, the AUC was significant but low powered in terms of sensitivity. Nevertheless, our study is the first to investigate fetuin-A and adiponectin levels in diabetic children, as compared to those in non-diabetic children. The significance of FAR in predicting metabolic complications among obese participants was highlighted. Data on fetuin-A and adiponectin levels in such populations could be valuable for future reference, considering that these data are scarce.

## 5. Conclusions

Plasma fetuin-A levels were significantly elevated in obese children and adolescents, and its association with IR and β cell function seemed to be more prominent in those with diabetes when adjusted for adiponectin. Therefore, FAR could be a useful marker for the early detection of metabolic complications in diabetic children and adolescents, particularly in those who are obese. Larger prospective studies are required to further investigate the determinants of plasma fetuin-A and adiponectin levels in association with childhood diabetes.

## Figures and Tables

**Figure 1 children-08-01155-g001:**
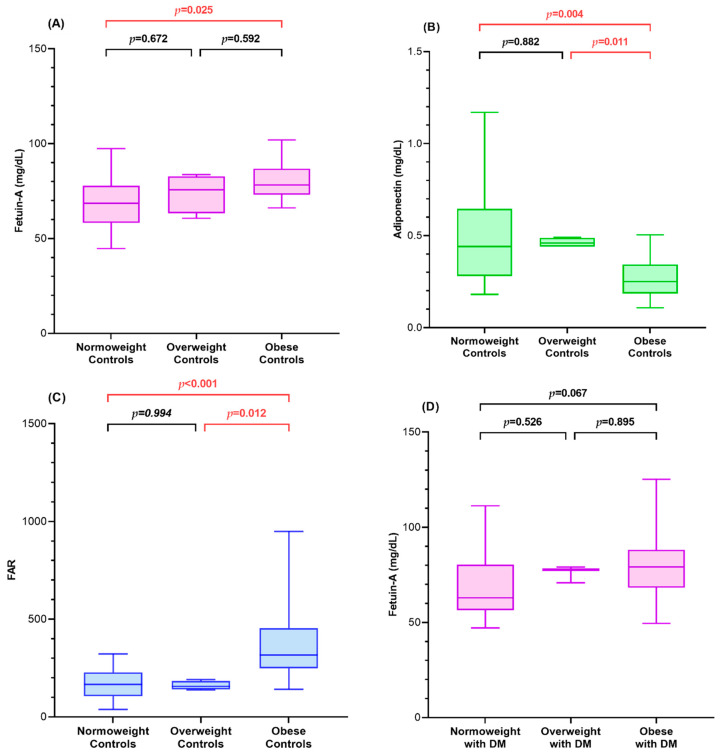
Controls (**A**–**C**) and diabetic children (**D**–**F**) were subcategorized into three groups according to body mass index status. Box plots were drawn to compare fetuin-A, adiponectin, and the fetuin-A-to-adiponectin ratio. Boxes represent the interquartile range, whereas lines inside the boxes represent the median value. Whiskers represent the lowest and highest observations.

**Figure 2 children-08-01155-g002:**
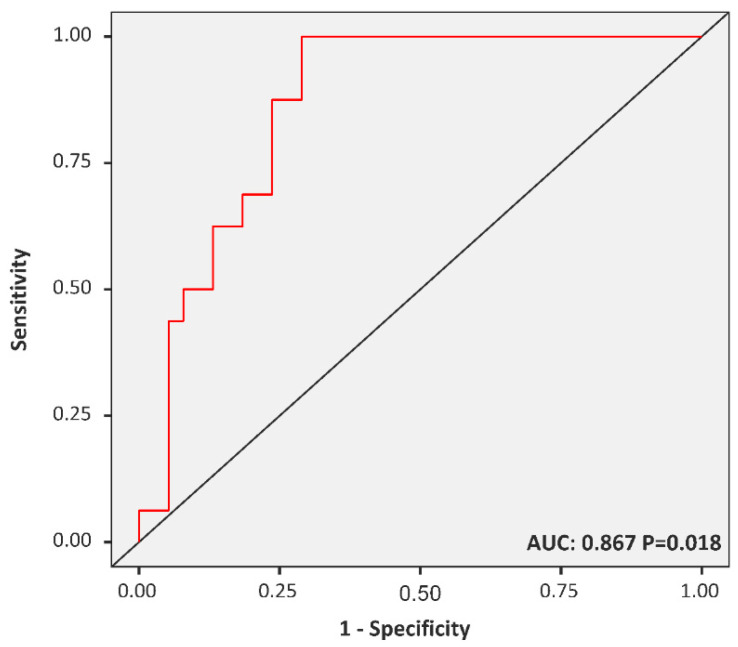
Receiver operating characteristic curve of the fetuin-A-to-adiponectin ratio for identifying children with metabolic syndrome. The optimal cut-off points and area under the curves (sensitivity, specificity) were 284.91 and 0.87 (0.25, 0.94) for diabetic children.

**Table 1 children-08-01155-t001:** Description of the study participants.

Clinical Parameters	Total (*n* = 98)	NOB (*n* = 45)	OB (*n* = 53)
Male, *n* (%)	55 (58.1)	21 (46.7)	34 (64.2)
Age, years	12.42 (9.56, 16.17)	12.08 (9.42, 16.04)	12.08 (9.92, 16.31)
Institution of enrollment			
Seoul St. Mary’s Hospital, *n* (%)	64 (65.3)	28 (62.2)	26 (49.1)
Bucheon St. Mary’s Hospital, *n* (%)	34 (34.7)	17 (37.8)	27 (50.9)
Diabetes mellitus, *n* (%)	54 (55.1)		27 (50.9)
Type 1, *n* (%)	31 (57.4)		7 (25.9)
Type 2, *n* (%)	23 (42.6)	27 (60.0)	20 (74.1)
Duration, years	1.58 (1.00, 4.13)	24 (88.9)	1.67 (1.25, 3.83)
MetS, *n* (%)	19 (19.4)	3 (11.1)	19 (35.8)
Controls, *n* (%)	3 (15.8)	1.5 (0.5, 4.5)	4 (21.1)
Type 1, *n* (%)	4 (21.1)	0 (0.0)	12(63.2)
Type 2, *n* (%)	12 (63.2)		3 (15.8)
Anthropometry			
Height SDS	0.43 ± 1.14	0.00 ± 1.01	0.80 ± 1.09
Weight SDS	1.42 ± 1.75	−0.17 ± 0.99	2.78 ± 0.86
BMI SDS	1.58 ± 1.95	−0.19 ± 1.10	3.09 ± 0.99

Parametric values are expressed as mean ± standard deviation, whereas non-parametric values are presented as median (interquartile range: 25%, 75%), unless otherwise stated. BMI, body mass index; MetS, metabolic syndrome; NOB, non-obese; OB, obese; SDS, standard deviation score.

**Table 2 children-08-01155-t002:** Clinical and biochemical characteristics of the study participants with respect to obesity and diabetes status.

	Total (*n* = 98)				*p*
	OC (*n* = 26)	NOC (*n* = 18)	ODM (*n* = 18)	NODM (*n* = 36)	
Age (years) ^2,3,4,5,6^	10.54 (9.42, 12.04)	8.92 (8.60, 10.48)	15.83 (15.00, 18.17)	14.67 (10.8, 16.58)	<0.001
BMI SDS ^1,3,4,6^	2.73 (2.46, 3.24)	0.24 (−0.56, 1.05)	3.03 (2.43, 3.99)	−0.62 (−1.44, 0.44)	<0.001
CRP (mg/dL) ^1,2,4,5,6^	0.10 (0.05, 0.19)	0.03 (0.03, 0.10)	0.87 (0.19, 2.60)	0.30 (0.03, 0.48)	<0.001
Glucose (mg/dL) ^2,3,4,5,6^	91 (91, 97.50)	96 (90, 97.50)	161 (113, 248)	110 (100, 128)	<0.001
AST (U/L) ^3,5,6^	22 (18, 28)	24 (20, 28.50)	27 (16, 59)	17 (13, 19)	<0.001
ALT (U/L) ^1,3,4,6^	18 (13, 41)	14 (9, 16.50)	54 (16, 103)	10 (9, 14)	<0.001
TC (mg/dL)	170 (156.50, 196)	179 (150.50, 222)	182 (159, 205)	170 (149, 199)	0.770
TG (mg/dL) ^2,6^	98 (56, 129.50)	96 (75, 150.50)	151 (109, 238)	78 (46, 116)	0.002
HDL-C (mg/dL) ^6^	49 (44, 55.50)	52 (44, 64.50)	45 (40, 55)	57 (50, 67)	0.003
LDL-C (mg/dL)	104 (95.50, 128)	108 (84.5, 131)	108.80 (81.40, 124)	96 (77.20, 127)	0.401
Uric acid (mg/dL) ^1,3,4,6^	6.35 (4.88, 7.13)	4.30 (3.80, 4.88)	5.40 (4.50, 6.30)	4.40 (3.50, 5.30)	<0.001
HbA1c (%) ^1,2,3,4,5^	5.50 (5.38, 5.70)	5.20 (5.08, 5.43)	8.80 (6.70, 10.30)	7.60 (6.70, 9.10)	<0.001
C-peptide (ng/mL) ^1,3,5,6^	2.63 (2.18, 3.79)	2.01 (1.53, 2.75)	2.56 (0.90, 4.21)	0.17 (0.02, 0.69)	<0.001
HOMA-IR ^3,4,5,6^	4.79 (2.73, 6.50)	2.46 (1.77, 4.94)	6.33 (3.51, 12.84)	0.84 (0.38, 1.48)	<0.001
HOMA-β ^1,2,3,5,6^	229.24 (171.40, 299.24)	124.65 (89.18, 225.73)	75.21 (26.54, 163.44)	26.02 (9.75, 50.40)	<0.001
Fetuin-A (mg/dL) ^1,3^	78.14 (73.06, 86.75)	70.16 (60.18, 80.93)	79.25 (68.27, 88.19)	65.52 (57.09, 80.20)	0.006
Adiponectin (mg/dL) ^1,3,4,6^	0.25 (0.18, 0.34)	0.45 (0.36, 0.52)	0.23 (0.13, 0.35)	0.56 (0.33, 0.91)	<0.001
FAR ^1,3,4,6^	316.38 (248.98, 454.17)	159.07 (123.38, 204.07)	341.56 (232.41, 653.07)	129.43 (60.79, 220.62)	<0.001

Values are expressed as median (interquartile range: 25%, 75%). ALT, alanine aminotransferase; AST, aspartate aminotransferase; BMI SDS, body mass index standard deviation score; CRP, C-reactive protein; FAR, fetuin-A-to-adiponectin ratio; HbA1c, glycosylated hemoglobin; HDL-C, high-density lipoprotein cholesterol; HOMA-IR, homeostatic model assessment of insulin resistance; HOMA-β, homeostatic model assessment of β cell function; LDL-C, low-density lipoprotein cholesterol; NOC, non-obese controls; NODM, non-obese diabetic children; OC, obese controls; ODM, obese diabetic children; TC, total cholesterol; TG, triglycerides. ^1^ Difference in *p*-value less than 0.05 between OC and NOC. ^2^ Difference in *p*-value less than 0.05 between OC and ODM. ^3^ Difference in *p*-value less than 0.05 between OC and NODM. ^4^ Difference in *p*-value less than 0.05 between NOC and ODM. ^5^ Difference in *p*-value less than 0.05 between NOC and NODM. ^6^ Difference in *p*-value less than 0.05 between ODM and NODM.

**Table 3 children-08-01155-t003:** Clinical and biochemical characteristics of the study participants with respect to the type of diabetes.

	Total (*n* = 98)			*p*
	Controls (*n* = 44)	T1DM (*n* = 31)	T2DM (*n* = 23)	
Age (years) ^1,2,3^	9.92 (8.67, 11.58)	14.67 (10.08, 16.42)	17.58 (15.75, 18.25)	<0.001
BMI SDS ^1,2,3^	2.22 (0.78, 2.92)	0.15 (−1.16, 1.59)	3.19 (1.99, 4.66)	<0.001
CRP (mg/dL) ^1,3^	0.07 (0.03, 0.14)	0.3 (0.03, 0.73)	0.87 (0.12, 2.60)	<0.001
Glucose (mg/dL) ^1,3^	94 (90.75, 97.25)	113 (101, 161)	122 (112, 219)	<0.001
AST (U/L) ^1^	22.5 (18.75, 28.25)	17 (14, 21)	34 (13, 59)	0.002
ALT (U/L) ^1,2,3^	15 (11.75, 22.5)	12 (9, 15)	57 (12, 103)	<0.001
TC (mg/dL)	173.5 (155.75, 203.75)	170 (149, 204)	181 (159, 202)	0.981
TG (mg/dL) ^3^	97 (58.75, 133.5)	79 (50, 156)	127 (99, 230)	0.025
HDL-C (mg/dL) ^1,2,3^	50 (44, 60.25)	59 (52, 71)	44 (39, 49)	<0.001
LDL-C (mg/dL)	106 (90.75, 127)	90.4 (74, 124)	116.6 (90, 123.2)	0.118
Uric acid (mg/dL)	5.25 (4.3, 6.85)	4.40 (3.8, 5.3)	5.4 (4.7, 6.4)	0.002
HbA1c (%) ^1,2^	5.4 (5.2, 5.6)	8.5 (6.8, 9.6)	7.9 (6.5, 10.3)	<0.001
C-peptide (ng/mL) ^1,3^	2.45 (1.83, 3.42)	0.13 (0.02, 0.66)	3.26 (1.96, 4.61)	<0.001
HOMA-IR ^1,2,3^	3.87 (2.26, 5.72)	0.95 (0.38, 1.82)	6.59 (4.27, 12.84)	<0.001
HOMA-β (%) ^1,2,3^	200.35 (136.19, 292.07)	22.76 (9.53, 42.45)	107.92 (60.04, 209.39)	<0.001
Fetuin-A (mg/dL)	75.59 (68.32, 83.62)	66.24 (59.99, 80.41)	79.25 (68.27, 84.53)	0.128
Adiponectin (mg/dL) ^1,2,3^	0.31 (0.21, 0.44)	0.56 (0.33, 0.94)	0.19 (0.13, 0.31)	<0.001
FAR ^1,2,3^	252.48 (166.87, 346.68)	129.43 (60.79, 245.54)	407.77 (243.57, 653.07)	<0.001

Values are expressed as median (interquartile range: 25%, 75%). ALT, alanine aminotransferase; AST, aspartate aminotransferase; BMI SDS, body mass index standard deviation score; CRP, C-reactive protein; FAR, fetuin-A-to-adiponectin ratio; HbA1c, glycosylated hemoglobin; HDL-C, high-density lipoprotein cholesterol; HOMA-IR, homeostatic model assessment of insulin resistance; HOMA-β, homeostatic model assessment of β cell function; LDL-C, low-density lipoprotein cholesterol; T1DM, type 1 diabetes mellitus; T2DM, type 2 diabetes mellitus; TC, total cholesterol; TG, triglycerides. ^1^ Difference in *p*-value less than 0.05 between controls and T1DM. ^2^ Difference in *p*-value less than 0.05 between controls and T2DM. ^3^ Difference in *p*-value less than 0.05 between T1DM and T2DM.

**Table 4 children-08-01155-t004:** Spearman’s rank correlation of plasma fetuin-A, adiponectin, and the fetuin-A-to-adiponectin ratio with the clinical and laboratory characteristics of diabetic children and controls.

	Diabetic Children (*n* = 54)						Controls (*n* = 44)					
	Fetuin-A		Adiponectin		FAR		Fetuin-A		Adiponectin		FAR	
	ρ	*p*	Ρ	*p*	ρ	*p*	ρ	*p*	ρ	*p*	ρ	*p*
BMI SDS	0.404	0.002	−0.649	<0.001	0.679	<0.001	0.437	0.003	−0.455	0.002	0.540	<0.001
AST	0.428	0.001	−0.297	0.029	0.381	0.004	−0.076	0.624	−0.031	0.843	−0.004	0.980
ALT	0.430	0.001	−0.496	<0.001	0.560	<0.001	0.099	0.523	−0.376	0.012	0.358	0.017
TC	0.121	0.385	0.031	0.826	0.008	0.953	0.083	0.592	−0.185	0.230	0.143	0.354
TG	0.247	0.072	−0.512	<0.001	0.517	<0.001	0.156	0.311	−0.130	0.400	0.119	0.442
HDL-C	−0.303	0.026	0.688	<0.001	−0.663	<0.001	−0.167	0.277	0.288	0.058	−0.300	0.048
LDL-C	0.162	0.241	−0.229	0.096	0.229	0.096	0.114	0.462	−0.292	0.054	0.250	0.102
Uric acid	0.112	0.421	−0.522	<0.001	0.493	<0.001	0.214	0.163	−0.577	<0.001	0.594	<0.001
HbA1c	0.113	0.415	0.140	0.313	−0.084	0.547	0.222	0.148	−0.464	0.002	0.453	0.002
C-peptide	0.246	0.072	−0.776	<0.001	0.745	<0.001	0.249	0.104	−0.512	<0.001	0.560	<0.001
HOMA-IR	0.395	0.003	−0.680	<0.001	0.699	<0.001	0.068	0.662	−0.462	0.002	0.452	0.002
HOMA-β	0.200	0.147	−0.581	<0.001	0.560	<0.001	0.045	0.772	−0.399	0.007	0.390	0.009

ALT, alanine aminotransferase; AST, aspartate aminotransferase; BMI SDS, body mass index standard deviation score; FAR, fetuin-A-to-adiponectin ratio; HbA1c, glycosylated hemoglobin; HDL-C, high-density lipoprotein cholesterol; HOMA-IR, homeostatic model assessment of insulin resistance; HOMA-β, homeostatic model assessment of β cell function; LDL-C, low-density lipoprotein cholesterol; TC, total cholesterol; TG, triglycerides.

**Table 5 children-08-01155-t005:** Univariate and multivariate regression analyses of factors associated with an increased fetuin-A-to-adiponectin ratio in diabetic children and controls.

Risk Factors	Diabetic Children (*n* = 54)				Controls (*n* = 44)			
	Univariate		Multivariate		Univariate		Multivariate	
	β	*p*	β	*p*	β	*p*	Β	*p*
Age	15.9	0.364	3.74	0.819	41.7	<0.001	31.7	0.002
Sex	173	0.126	180.74	0.080	95.3	0.098	−24.3	0.635
BMI SDS	86.7	<0.001	85.57	<0.001	63.0	<0.001	49.6	0.004
AST	5.69	0.027			3.39	0.004		
ALT	3.49	0.006			1.66	0.001		
TC	−2.02	0.275			0.695	0.445		
TG	0.97	0.125			0.613	0.093		
HDL-C	−12.50	0.002			−4.75	0.065		
LDL-C	−0.31	0.881			0.908	0.354		
Uric acid	160.00	<0.001			65.3	<0.001		
HbA1c	−1.23	0.963			45.1	0.251		
C-peptide	115	<0.001			54.7	0.011		
HOMA-IR	7.13	0.169			4.78	0.356		
HOMA-β	1.05	0.070			0.276	0.169		

ALT, alanine aminotransferase; AST, aspartate aminotransferase; BMI SDS, body mass index standard deviation score; HbA1c, glycosylated hemoglobin; HDL-C, high-density lipoprotein cholesterol; HOMA-IR, homeostatic model assessment of insulin resistance; HOMA-β, homeostatic model assessment of β cell function; LDL-C, low-density lipoprotein cholesterol; TC, total cholesterol; TG, triglycerides.

## Data Availability

The raw data could be provided under authors’ permission without undue reservation.

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
