# Peer review of "Clinical Significance of the Fetuin-A-to-Adiponectin Ratio in Obese Children and Adolescents with Diabetes Mellitus"

_children, 2021, doi:10.3390/children8121155_

Round 1
Reviewer 1 Report
This study of Moon Bae Ahn et al. observed the modulated plasma fetuin-A and adiponectin levels in diabetic children and adolescents in relation to obesity. I'm the study is correctly ideated.
I am curious to know how this markers modulates in other population. for example adults with the same characteristics of the population observed in this study.
Author Response
Thank you for your thoughtful comments. Please find the attached file (.pdf) for author's notes to reviewer.

Reviewer 2 Report
Thank you for the opportunity to review this manuscript. This is an interesting topic that can be considered by readers. One minor comment - in my opinion, it is necessary to describe what specific results the authors expect (a hypothesis should be made in the introduction).
Author Response
Thank you for your thoughtful comments.
"Thank you for the opportunity to review this manuscript. This is an interesting topic that can be considered by readers. One minor comment - in my opinion, it is necessary to describe what specific results the authors expect (a hypothesis should be made in the introduction)."
Response:
Thank you for your thoughtful comments. We agree with your point that there lacks a hypothesis in Introduction section, therefore rephrased the text.
Contents rephrased (Line 64):
Although it has been hypothesized that the FAR could be a sensitive indicator of obese children and adolescents so as in those who are diabetic, clinical evidence regarding the value of fetuin-A, adiponectin, and FAR in such population is limited.
We appreciate for your comments.